# Fault Diagnosis of Wind Turbine Generators Based on Stacking Integration Algorithm and Adaptive Threshold

**DOI:** 10.3390/s23136198

**Published:** 2023-07-06

**Authors:** Zhanjun Tang, Xiaobing Shi, Huayu Zou, Yuting Zhu, Yushi Yang, Yajia Zhang, Jianfeng He

**Affiliations:** 1Faculty of Information Engineering and Automation, Kunming University of Science and Technology, Kunming 650500, China; sxb@stu.kust.edu.cn (X.S.); zouhy1993@vip.163.com (H.Z.); zyt@163.com (Y.Z.); yangyushi@stu.kust.edu.cn (Y.Y.); jfenghe@kust.edu.cn (J.H.); 2Key Laboratory of Artificial Intelligence in Yunnan Province, Kunming 650500, China; 3Yunnan Open University, Kunming 650500, China; star@stu.kust.edu.cn

**Keywords:** fault diagnosis, WTG, stacking integration algorithm, machine learning, EDP, adaptive threshold

## Abstract

Fault alarm time lag is one of the difficulties in fault diagnosis of wind turbine generators (WTGs), and the existing methods are insufficient to achieve accurate and rapid fault diagnosis of WTGs, and the operation and maintenance costs of WTGs are too high. To invent a new method for fast and accurate fault diagnosis of WTGs, this study constructs a stacking integration model based on the machine learning algorithms light gradient boosting machine (LightGBM), extreme gradient boosting (XGBoost), and stochastic gradient descent regressor (SGDRegressor) using publicly available datasets from Energias De Portugal (EDP). This model is automatically tuned for hyperparameters during training using Bayesian tuning, and the coefficient of determination (R^2^) and root mean square error (RMSE) were used to evaluate the model to determine its applicability and accuracy. The fitted residuals of the test set were calculated, the Pauta criterion (3σ) and the temporal sliding window were applied, and a final adaptive threshold method for accurate fault diagnosis and alarming was created. The model validation results show that the adaptive threshold method proposed in this study is better than the fixed threshold for diagnosis, and the alarm times for the GENERATOR fault type, GENERATOR_BEARING fault type, and TRANSFORMER fault type are 1.5 h, 5.8 h, and 3 h earlier, respectively.

## 1. Introduction

As the global demand for energy continues to climb, the massive consumption of fossil-fuel-derived energy not only leads to increased environmental pollution, but also limits the sustainable socio-economic development. Therefore, renewable energy has been developed rapidly and has gradually become a research hotspot, among which wind energy is one of the most important components of renewable energy [1,2,3] and plays a key role in the low-carbon transformation of the power industry. Currently, the global wind power industry is growing rapidly and the cumulative installed capacity is gradually climbing [4]. According to the World Wind Energy Report 2023 published by the Global Wind Energy Council (GWEC) [5], the global installed wind power capacity was 77.6 GW in 2022 and is expected to exceed 100 GW in 2024. The data shows that wind power is one of the most important energy sources, therefore, parameters such as wind power energy efficiency, wind turbine fault diagnosis, and fault warning forecasting should be given more attention.

However, traditional wind power plants rely on regular maintenance and post maintenance, and the long deployment cycle of spare parts leads to high failure maintenance costs, which has a huge impact on the operation and maintenance economics of wind power plants [6]. In addition, wind turbine generators (WTGs) are usually installed in areas with abundant wind energy resources but that are remote and harsh environments, making equipment maintenance more difficult [7]. The aging of WTGs leads to the degradation of power production performance and reduces the economic efficiency. Lagging fault diagnosis can lead to faults not being detected in time, which in turn delays fault repair and maintenance. In addition, the continued operation of WTGs under fault conditions increases the load and risk, compromises the reliability and performance of the equipment, and leads to high O&M costs for WTGs. Therefore, to effectively reduce downtime and avoid serious component damage, there is an urgent need to explore an efficient fault diagnosis technique to achieve an efficient and sustainable diagnosis method for wind turbines. This method can optimize the maintenance strategy and avoid unnecessary maintenance activities, so that the fault diagnosis technique can become a way to reduce the O&M cost of wind power projects and finally maximize the profit [8,9,10].

Existing methods for fault diagnosis of WTGs can be divided into model-based methods, vibration signal methods, and data-driven methods [11,12,13]. Model-based methods usually require a mathematical model to describe the structure, characteristics, and operation of the WTG, and ultimately detect potential faults and abnormalities through simulation and analysis [14]. For example, Dey et al. [15] proposed a cascade approach based on two Kalman filters to construct a detection model that mitigates the effect of nonlinear aerodynamic torques in the drivetrain dynamics. Jlassi et al. [16] proposed a model based on a doubly-fed wind power system, where a state observer of the rotor current was established to finally achieve the detection of rotor converter faults. However, the model building process is complex and tedious, requiring consideration of multiple factors and parameters, and the accuracy of models is affected by various factors such as parameter uncertainties and external environmental disturbances, thus affecting their diagnostic veracity and accuracy [17]. The vibration signal method is a common non-destructive testing (NDT) technique that can detect faults and abnormalities that are present in a unit by collecting, processing, and analyzing vibration signals without damaging the object under test, such as bearing faults, gear faults, and blade faults [18,19,20,21,22]. For example, Rezamand et al. [23] created a feature-based fault prediction method in the process of the fault study of wind turbine bearings. Although certain results have been achieved in the failure of bearings, the reasonableness of the feature selection for this study needs to be considered. However, the data of the vibration signal method are susceptible to interference from environmental noise, leading to misclassification or omission, which limits the prospects of this method in fault diagnosis [24,25]. Data-driven methods are mainly used for fault diagnosis using the supervisory control and data acquisition (SCADA) data system, and this system is capable of collecting rich operational data and operating for all available conditions, such as wind speed, temperature, and rotational speed. This system can monitor the operating status of the equipment in real time and can issue fault alerts in time. In addition, the system can communicate with other devices to exchange data and take measures to protect the system from malicious intrusion [2,26,27,28,29,30,31]. Wang et al. [32] proposed a deep confidence network for feature learning and classification based on SCADA data to achieve sensor fault detection. Among them, neural networks (NNs), support vector machine (SVM), random forest (RF), and k-nearest neighbors (KNN) [33,34,35,36,37,38,39,40] are the most commonly used algorithms. However, this method has high data requirements, and the handling of missing data and outliers needs careful consideration. In summary, most studies on wind turbine fault diagnosis still use a single algorithmic model. Due to the limitations of the models themselves, these methods usually have difficulty in capturing complex fault patterns and are susceptible to outliers or noise [41]. Therefore, the use of integrated learning is important to improve the accuracy and robustness of classifiers. Stacking integration algorithms can well capture the specific difficulties between different algorithms and weigh the advantages between them to improve the accuracy and reliability of fault diagnosis [42]. The use of a stacking integrated algorithm model has certain advantages for accurate fault diagnosis in WTGs.

Therefore, a fault diagnosis method for WTGs based on a stacking integration algorithm and adaptive thresholds was proposed. In this study, a fusion model of a stacking integrated algorithm was used to integrate the light gradient boosting machine (LightGBM), extreme gradient boosting (XGBoost), and stochastic gradient descent regressor (SGDRegressor) algorithms. In addition, this study also uses the Bayesian tuning method to tune the model parameters, thus significantly improving the comprehensive diagnostic performance of the model. At the level of the diagnosis method, this study proposes an adaptive threshold method combining the Pauta criterion (3σ) and a temporal sliding window, which can effectively reduce the false alarm situations during the diagnosis process and realize the diagnosis and alarm of the fault in advance. This diagnosis method for WTGs proposed in this study makes full use of the advantages of linear algorithms and decision tree algorithms, while making up for the shortcomings of each, so that it has better information extraction and fault analysis capabilities. In addition, this study creates an adaptive threshold method for fault diagnosis, which provides a new perspective for the accurate diagnosis of wind turbine faults and has obvious innovation and value, and can provide technical support for the development of the global wind power business.

## 2. Methods and Principles

The LightGBM and XGBoost were chosen as the base learners, and, finally, the SGDRegressor was selected as the meta-learner, to construct the stacking integrated model. Meanwhile, the hyperparameters were tuned using the Bayesian tuning method. The prediction effectiveness of the model was evaluated using the coefficient of determination (R^2^) and root mean square error (RMSE). In addition, an adaptive threshold fault precision diagnosis and alarm method was proposed by combining the 3σ criterion and the temporal sliding window, where the principles of each method are described in the following sections.

### 2.1. LightGBM Algorithm

LightGBM, a machine learning algorithm with the gradient boosting decision tree (GBDT) algorithm, is a distributed gradient boosting framework based on a decision tree algorithm [43], and iterative learning with a reduced loss function. During the iterative computation, a weak learner is added to the strong learner in the previous learning process to make the updated strong learner with the minimum loss function, and the updated strong learner is as follows.
(1)$$F_{t}\left(x\right)=h_{t}\left(x\right)+F_{t-1}\left(x\right).$$

The weak learner expression is as follows.
(2)$$h_{t}\left(x\right)=\arg\underset{h\in H}{\min}\sum Loss\left(y,F_{t-1}\left(x\right)+h_{x}\right).$$

The individual sample’s negative gradients are calculated as approximate residuals each time the learner is updated.
(3)$$r_{ti}=-\frac{\partial Loss\left(y,F_{t-1}\left(x_{i}\right)\right)}{\partial F_{t-1}\left(x_{i}\right)}.$$

The $h_{t}\left(x\right)$ after fitting with the squared difference is as follows.
(4)$$h_{t}\left(x\right)=\arg\underset{h\in H}{\min}{\sum \left(r_{ti}-h\left(x\right)\right)}^{2},$$
where $h_{t}\left(x\right)$ is the updated weak learner, $F_{t-1}\left(x\right)$ is the strong learner before the update, H is the set consisting of all possible base learners, and argmin is the minimum value for solving the loss function.

### 2.2. XGBoost Algorithm

XGBoost is an optimized distributed gradient enhancement library and an integrated learning model based on decision trees [44]. This model can improve the calculation of the objective function based on gradient boosting, which transforms the optimization problem of the objective function into the minimization problem of the quadratic function and uses the information of the second-order derivatives of the loss function to train the tree model to improve the accuracy of the model. At the same time, the generalization performance of the model is improved by adding the complexity of the tree as a regularization term to the objective function. The objective function of XGBoost is calculated as follows.
(5)$$J=\sum_{i=1}^{n}L\left(y_{i},\overset{\wedge }{y_{i}}\right)+\sum_{k=1}^{K}\Omega \left(f_{k}\right).$$

The results of the iterations of the objective function in time period t are as follows.
(6)$$J^{\left(t\right)}\approx \sum_{i=1}^{n}[L\left(y_{i},{\overset{\wedge }{y_{i}}}^{t-1}\right)+f_{t}\left(x_{i}\right)]+\Omega \left(f_{t}\right)+C.$$

The loss function is subjected to a second-order Taylor expansion, and then the objective function is calculated as follows.
(7)$$J^{\left(t\right)}\approx \sum_{i=1}^{n}[L\left(y_{i},{\overset{\wedge }{y_{i}}}^{t-1}\right)+g_{i}f_{t}\left(x_{i}\right)+\frac{1}{2}h_{i}f_{t}{}^{2}\left(x_{i}\right)]+\Omega \left(f_{t}\right)+C,$$
where $y_{i}$ is the actual value of the i-th target, $\overset{\wedge }{y_{i}}$ is the predicted value of the i-th target, $L\left(y_{i},\overset{\wedge }{y_{i}}\right)$ is the difference between $y_{i}$ and $\overset{\wedge }{y_{i}}$, n is the sample size, $\Omega \left(f_{k}\right)$ is the tree complexity, K is the number of sample features, $f_{t}\left(x_{i}\right)$ is the complexity of the decision tree where the variable $x_{i}$ is located, C is a constant, and $g_{i}$ and $h_{i}$ are the first-order and second-order derivatives of the loss function, respectively.

### 2.3. SGDRegressor Algorithm

SGDRegressor is a regression algorithm that uses stochastic gradient descent (SGD) to minimize the loss function and is suitable for handling large-scale datasets. Compared with the traditional batch gradient descent, SGDRegressor approximates the $E\left(w,b\right)$ true gradient by considering only one training sample per iteration, resulting in an efficient training process [45]. Due to the stochastic nature of the decline, SGDRegressor requires more iterations to reach the minimum, but it is still computationally inexpensive. Least squares are used as a loss function to calculate the total squared error between the actual data and the associated hyperplane. L2 regularization can improve the generalization performance by applying a square penalty to the weight vector of the model and limiting the complexity of the model. To improve the generalization ability of the model and avoid overfitting, the L2 regularization term was introduced into the model. In this study, the combination of the loss function and L2 regularization term was applied, thus balancing the fit of the model on the training data, improving the generalization ability to unseen data, and effectively reducing the overfitting problem. The loss function and the regularization term are used to fit the regression model with the following equations.
(8)$$E\left(w,b\right)=\frac{1}{n}\sum_{i=1}^{n}L\left(y_{i},f\left(x_{i}\right)\right)+\alpha R\left(w\right),$$
(9)$$L\left(y_{i},f\left(x_{i}\right)\right)=\frac{1}{2}{\left(y_{i}-f\left(x_{i}\right)\right)}^{2},$$
(10)$$R\left(w\right)=\frac{1}{2}\sum_{j=1}^{m}w_{j}{}^{2}={\left\|w\right\|}_{2}^{2}$$
where L is the loss function, R is the regularization term (L2 parametric), and α>0 hyperparameters are used to control the regularization strength.

### 2.4. Stacking Integration Algorithm

Ensemble learning is an approach to complete a learning task by constructing and combining multiple learners. It first generates a set of base learners and then combines them through some learning strategy to output the final result [46]. Currently, ensemble learning mainly includes homogeneous integration and heterogeneous integration. Bagging [47,48,49] and boosting [44] are the most classical homogeneous integration methods. Homogeneous integration patterns are highly correlated and prone to overfitting where the model over-fits the training dataset. Overfitting can lead to a decrease in the generalization ability of the model to the point where it does not generalize well to new, unseen data. In addition, frequent application of averaging can lead to a homogeneous integration model that simply linearizes the results of the base learner, from which it is difficult to accurately reflect the true diagnostic results. Heterogeneous integration primarily includes stacking [50]. The stacking algorithm is a typical representative of heterogeneous integration methods and consists of two main layers of structure. Layer 1 is the base learner, which is used to train and predict the original samples. Layer 2 is the meta-learner, which is used to combine the diagnosis results of layer 1, to learn again, and perform the diagnosis. The high diversity of heterogeneous integration methods has significant algorithmic advantages, thus outperforming the prediction performance of a single learner [51]. In this study, LightGBM and XGBoost were used as base learners, and SGDRegressor was used as a meta-learner, combined with a 5-fold cross-validation method to train each base learner separately for integration, as shown in Figure 1.

(1) The original training set is divided into 5 subsets, denoted as D1–D5;

(2) In the 1st cross-validation, D2–D5 are input to the base learner as the training set, and the trained base model is denoted as B1. D1 is input to B1 as the validation set D, and the output is denoted as r1.

(3) This process is repeated. After reaching 5-fold cross-validation, 5 models (B1, B2, B3, B4, B5) are trained, and 5 results (r1, r2, r3, r4, r5) are output, and the output results are combined into a set and then averaged (R1).

(4) The test set is input into B1–B5, and the output results are recorded as n1–n5, and after taking the average value, they are recorded as N1.

(5) Finally, after the above 4 steps are executed for both of the 2 base models, the diagnostic results R1 and R2 and the mean values N1 and N2 of the prediction results are output. These two parts of the results are used as the training set and test set of the second layer model, respectively, and the final results are output.

### 2.5. Bayesian Tuning

For the model, a suitable hyperparameter configuration can greatly improve its performance. Commonly used tuning methods include the manual tuning method, random search method [52], and Bayesian tuning method [53]. The manual tuning method cannot find the best combination of parameters and is time consuming. The random search method does not make full use of the a priori information. In contrast, the Bayesian tuning method can make full use of prior information to efficiently search for the optimal solution of hyperparameters in fewer iterations. Its main steps include setting the parameter search range, randomly generating the initial set of candidate solutions, finding the next point based on these points, and repeating this step until the end of the iteration. Therefore, the Bayesian tuning method is chosen to automatically tune the parameters of the algorithm in the model training process of this study.

The core of Bayesian optimization is the probabilistic agent model and the collection function. The probabilistic agent model is commonly used in Gaussian process regression, and the collection function is commonly used in probability of improvement (PI) [54]. The main idea is to use the posterior model to estimate the probability that the function value of an unknown point is greater than the existing maximum value, and take the point with the highest probability $X^{*}$ as the next search point. The collection function formula is shown below.
(11)$$PI\left(X^{*}\right)=P\left(f\left(x\right)\geq f\left(x^{+}\right)+\theta \right)=\phi \left(\frac{\mu \left(x\right)-f\left(x^{+}\right)-\theta }{\sigma \left(x\right)}\right),$$
where $X^{*}$ is the next search point, θ>0 denotes the equilibrium parameter for exploring unknown points or exploring nearby maxima using known points, $\phi \left(.\right)$ denotes the cumulative density function of the standard normal cumulative distribution, $f\left(x\right)$ is the target value, $f\left(x^{+}\right)$ denotes the existing maximum value, $\mu \left(x\right)$ represents the mean of the objective function of the Gaussian process, and $\sigma \left(x\right)$ represents its standard deviation.

### 2.6. Evaluation of the Model

The coefficient of determination (R^2^) is a commonly used regression model predictor to measure the correlation between model predictions and actual results. The value of R^2^ ranges from [0, 1], and usually the closer the value is to 1, the better the predictive ability of the model, and vice versa [55].
(12)$$R^{2}=\frac{\sum_{i=1}^{n}{\left(\overset{\wedge }{y_{i}}-\overset{-}{y}\right)}^{2}}{\sum_{i=1}^{n}{\left(y_{i}-\overset{-}{y}\right)}^{2}}$$
where $y_{i}$ denotes the true value, $\overset{\wedge }{y_{i}}$ denotes the predicted value of the model, and $\overset{-}{y}$ denotes the mean value.

The root mean squared error (RMSE) is a commonly used metric to assess the predictive performance of regression models and is interpretable to penalize the squared loss of large deviations. A smaller value of RMSE indicates a smaller degree of difference between the predicted and true values, thus indicating a better prediction of the model [56]. The RMSE is calculated by the following formula.
(13)$$RMSE=\sqrt{\frac{1}{N}\sum_{i=1}^{N}{\left(y_{i}-\overset{\wedge }{y_{i}}\right)}^{2}}$$
where N denotes the number of samples, $y_{i}$ denotes the true value, and $\overset{\wedge }{y_{i}}$ denotes the predicted value of the model.

### 2.7. Pauta Criterion (3σ)

The Pauta criterion is a statistical method used to assess the dispersion and stability of a dataset and can help to identify possible outliers. According to the 3σ criterion, the probability that a data point is within the interval (μ − σ, μ + σ) is 0.6827, the probability that a data point is within the interval (μ − 2σ, μ + 2σ) is 0.9544, and the probability that a data point is within the interval (μ − 3σ, μ + 3σ) is 0.9974 [57]. Data points outside the range indicate that there are possible outliers in these data points that require further examination and analysis. In the fault diagnosis process of this study, the 3σ criterion calculates specific thresholds, and then an adaptive threshold method is proposed by combining the idea of time-series sliding windows.

## 3. Modeling Methods for Troubleshooting

Firstly, the SCADA data were cleaned, and then the features were filtered by the Pearson correlation coefficient, Spearman correlation coefficient, and variance selection, and the LightGBM, XGBoost, and SGDRegressor models were constructed separately. Finally, the stacking integrated model was constructed. This integrated model was applied to predict the potential fault data by multi-objective regression and calculate the fitted residuals to realize the fault diagnosis of multiple components of wind turbines; the modeling process is shown in Figure 2.

### 3.1. Source of Data

The dataset was obtained from the Energias De Portugal (EDP) open data platform [58]. This dataset contains all the data of five wind turbines (T01, T06, T07, T09, T11), each with a rated power of 2 MW, whose theoretical cut-in wind speed is 4 m/s and the theoretical rated wind speed is 12 m/s. The generator bearing temperature, oil temperature, pitch angle, generator speed, and other types of data were recorded by the sensors. The data were recorded at a frequency of every 10 min and the data spanned from 2016 to 2017, containing a total of 521,784 data with 83 characteristic columns. Some of the data are shown in Table 1. The data from the T06 and T07 units, including 102,922 and 104,739 records, respectively, were selected for this study.

In addition, the fault logging data includes 5 different types of faults (28 in total), namely, GEARBOX, GENERATOR, GENERATOR_BEARING, HYDRAULIC_GROUP, and TRANSFORMER. The data used in this study are mainly temperature data, and the temperature data were collected by temperature sensors. Temperature sensors are used to monitor the temperature of individual components to ensure proper system operation and reduce thermal damage to critical components. There are specific requirements for the installation location of temperature sensors. For example, generator temperature sensors are usually installed on the stator and rotor to monitor the generator temperature; gearbox temperature sensors are usually installed internally to monitor gear and bearing temperatures; and transformer temperature sensors are usually installed on the heat sink or critical components to ensure proper operating temperatures. However, not all types of faults can be reflected by temperature changes. Therefore, GENERATOR, GENERATOR_BEARING, and TRANSFORMER-type fault data should be used as the focus of the study for fault diagnosis and are shown in Table 2.

### 3.2. Cleaning of Data

As shown in Table 3, there are a few missing values in some feature columns in the dataset. To solve this problem, a forward interpolation method is used to fill in these missing values. When the wind turbine is shut down, the data recorded by the SCADA system may not reflect the actual generator status and performance. For example, the value of the “Grd_Prod_PsblePwr_Avg” characteristic column has 0 or negative values, which does not match the actual situation. Therefore, in order to ensure the accuracy of the data, these invalid data need to be filtered and eliminated when performing troubleshooting or performance evaluation, in order to avoid the impact on subsequent analysis and research. The aim is to improve the reliability of the analysis results and to ensure that studies and applications based on the SCADA data more accurately reflect the true operating status of WTGs.

### 3.3. Filtering of Features

Three metrics, the Pearson correlation coefficient, Spearman correlation coefficient, and variance, were used to screen the characteristics. Among them, the Pearson correlation coefficient is used to measure the linear correlation between two variables, which is obtained by dividing the covariance by the standard deviation of the two variables. Although the covariance can reflect the correlation degree of two random variables, its value is greatly influenced by the magnitude, and the judgment of the correlation degree of variables cannot be simply given from the numerical size of the covariance. To eliminate the effect of this magnitude, the concept of the correlation coefficient is introduced. The correlation coefficient is meaningful when the variance of both variables is not zero, and the correlation coefficient takes values in the range [−1, 1]. Usually, when the absolute values of the characteristic variable and the target variable in the correlation coefficient are less than 0.2, the correlation can be considered as very low and directly excluded [59]. The Spearman correlation coefficient measures the nonlinear relationship between two variables and is a nonparametric measure of rank correlation, which can be viewed as a nonparametric version of the Pearson correlation. What is observed is the strength of the monotonic relationship between the two, which, in layman’s terms, is the extent to which the two keep in step with each other in terms of their tendency to become larger or smaller. When the absolute value of the characteristic variable and the target variable in the correlation coefficient is less than 0.2, the correlation can be considered low and directly excluded [60]. The variance selection method first calculates the variance of each feature, and then selects the features with variance greater than the threshold value according to the threshold value, and screens the features with a very high proportion of single values. Because when a feature does not diverge, for example, the variance is close to 0, which means that the sample basically does not differ in this feature, this feature is basically meaningless for the differentiation of the sample. Generally, when the variance is less than 0.1, it can be considered as an invalid feature and directly rejected [61]. The application of these methods helps to improve the accuracy and validity of feature selection and ensure that the selected features can better reflect the characteristics of the study object.

The model construction work for different fault types was processed relatively independently, so the feature screening process was also relatively independent. After feature screening, 70 features were retained and 15.66% of the features were rejected for the GENERATOR-type of unit T06. The GENERATOR_BEARING-type of the T07 unit retains 66 features and rejects 20.48% of the features; the TRANSFORMER-type of the T07 unit retains 63 features and rejects 24.10% of the features. To ensure that features are selected to meet the modeling needs and high-performance requirements for different fault types, multiple target feature columns are also selected for each fault type in the SCADA data, as shown in Table 4.

### 3.4. Division of Data

Based on the fault advance diagnosis time defined by EDP, this study defines the data before 60 days from the occurrence of the fault as the normal operation data of the WTGs and sets their label as 0. The data 60 days before the occurrence of the fault is defined as the potential fault data of the WTGs and their label is set as 1. The interquartile range (IQR) criterion was used to filter the target characteristics from the normal operating data to reduce the interference of outliers, thus improving the accuracy and robustness of the model.

To meet the requirements, the datasets divided under the GENERATOR fault type for unit T06, GENERATOR_BEARING fault type for unit T07, and TRANSFORMER fault type for unit T07 were used to construct independent multi-objective regression models. The SCADA data in each unit were divided into three parts: training set, validation set and test set. Among them, the normal operation data were randomly divided into training and validation sets in the ratio of 8:2, and the potential fault data were set as the test set, as shown in Table 5. Each unit corresponds to a different fault type to ensure the generalization performance and reliability of the model.

### 3.5. Construction of the Model

Python 3.8 and its third-party libraries were used to implement the code runs and modeling. LightGBM, XGBoost, and SGDRegressor were integrated by stacking integrated algorithms and multi-objective regression models were built for the diagnosis of each fault type. To optimize the model performance, Bayesian tuning was applied and parameter tuning was performed in the form of five-fold cross-validation, with the number of calculations for the optimal parameter set to 30 to ensure better performance and reliability of the model.

In the training process of the LightGBM single model, the max_depth, learning_rate, and num_leaves parameters were tuned to find their optimal values. In the training process of the XGBoost single model, max_depth, min_child_weight, learning_rate, gamma, and subsample parameters were tuned to find their optimal values. In the training process of the SGDRegressor single model, the parameters of the l1_ratio were tuned to find their optimal values. Meanwhile, to eliminate the interference of different magnitudes on the linear model training, the MinMaxScaler method was employed to normalize the entire data, and each feature was scaled to between 0 and 1. After completing the training of the three single models, the stacking integration framework was used, with LightGBM and XGBoost as the base learners and SGDRegressor as the meta-learner, to train the integrated model in the form of five-fold cross-validation.

The results of the R^2^ and RMSE evaluations of the multi-objective regression models for different fault types are detailed in Table 6, Table 7 and Table 8. Since the test set is potential fault data, the prediction effect is supposed to be poor, and its evaluation value should be significantly lower than that of the training and validation sets. The evaluation results of the validation set show that the evaluated values of the four models corresponding to each fault type are not significantly different between the training and validation sets, indicating that the overfitting degree of all models is small and meets the expected objectives. The stacking integrated model has the best R^2^ and RMSE results on the validation set, indicating that the stacking integrated model has better prediction results compared to the single model.

## 4. Troubleshooting of Wind Turbines

The method of setting fault diagnosis thresholds for temperature varies in different application scenarios and detection objects. Generally, fixed thresholds are obtained by calculating the mean and standard deviation of temperature data over a period of time based on the 3σ criterion. However, with large temperature variations caused by the working environment, the fixed-threshold method will not be able to make adaptive adjustments, which can easily lead to false alarms. This study proposes an adaptive thresholding method that fully integrates the idea of the 3σ criterion and time-series sliding window to calculate the mean value of residuals within each window, and the standard deviation of residuals is obtained by the validation set calculation. The residual standard deviation is then obtained by subtracting the residual mean from the validation set for each sliding window.

The fitted residual curves for the three temperature measurement points corresponding to GENERATOR (Gen_Phase1_Temp_Avg, Gen_Phase2_Temp_Avg, and Gen_Phase3_Temp_Avg) are shown in Figure 3. By fitting the residual data, a fixed threshold value of 14.0701 is calculated. Obviously, the residual data of the validation set does not exceed this threshold value and no alarm phenomenon occurs. The fitted residual curves for the three temperature measurement points (Gen_Phase1_Temp_Avg, Gen_Phase2_Temp_Avg, Gen_Phase3_Temp_Avg) of the GENERATOR test set are shown in Figure 4. Among them, compared with the other two measurement points, the fitted residuals of the measurement point “Gen_Phase2_Temp_Avg” increased abnormally at around 10:00 pm on 23 July 2016. Based on this abnormal phenomenon, troubleshooting was performed for the abnormal time period of this measurement point.

As shown in Table 2, unit T06 has a GENERATOR-type fault recorded at time point “24 July 2016 17:01”. The fault diagnosis result of “Gen_Phase2_Temp_Avg” is shown in Figure 5. The residuals (blue curve), fixed-threshold line (gray horizontal line), adaptive-threshold curve (dark red curve), fault logging time point (blue vertical line), fixed-threshold alarm time point (gray vertical line), and adaptive-threshold alarm time point (dark red vertical line) for the “Gen_Phase2_Temp_Avg” measurement point are displayed. In the fault diagnosis for the abnormal time period of this measurement point, the fixed-threshold alarm time point is “23 July 2016 23:00” and the adaptive-threshold alarm time point is “23 July 2016 21:30”. Both threshold alarm time points are earlier than the fault record point “24 July 2016 17:01”, and the adaptive-threshold alarm time is 1.5 h earlier than the fixed-threshold alarm time.

The fitted residual curves for the two temperature measurement points (Gen_Bear_Temp_Avg and Gen_Bear2_Temp_Avg) corresponding to the T07 unit in the GENERATOR_BEARING validation set are shown in Figure 6. A fixed threshold value of 14.7722 was calculated from the fitted residual data. The residual data in the validation set did not exceed this threshold value and no alarm occurred. The fitted residual curves of the two temperature measurement points (Gen_Bear_Temp_Avg and Gen_Bear2_Temp_Avg) corresponding to the GENERATOR_BEARING test set of unit T07 are shown in Figure 7. Among them, the fitted residuals of “Gen_Bear2_Temp_Avg” show a significant abnormal fluctuation at the time point “16 August 2017 14:00”, which is significantly different from another measurement point. Troubleshooting was performed for the abnormal time period of this measurement point.

As shown in Table 2, unit T07 has a fault record of GENERATOR_BEARING-type at time point “20 August 2017 06:08”. The fault diagnosis result of “Gen_Bear2_Temp_Avg” is shown in Figure 8. In the fault diagnosis of the abnormal time of this measurement point, the fixed-threshold alarm time point is “16 August 2017 20:50” and the adaptive-threshold alarm time point is “16 August 2017 15:00”. Both threshold alarm points are earlier than the fault record point “20 August 2017 6:08”, and the adaptive-threshold alarm time is about 5.8 h earlier than the fixed-threshold alarm time.

The fitted residual curves for unit T07 at the three temperature measurement points corresponding to the TRANSFORMER validation set (HVTrafo_Phase1_Temp_Avg, HVTrafo_Phase2_Temp_Avg, HVTrafo_Phase3_Temp_Avg) are shown in Figure 9. The fixed threshold value of 21.5524 is calculated from the fitted residual data. It can be seen that the residual data of the validation set does not exceed this threshold value and no alarm phenomenon occurs. The fitted residual curves of unit T07 for the three temperature measurement points corresponding to the TRANSFORMER test set (HVTrafo_Phase1_Temp_Avg,HVTrafo_Phase2_Temp_Avg,HVTrafo_Phase3_Temp_Avg) are shown in Figure 10. Among them, the fitted residuals of the measurement point “HVTrafo_Phase3_Temp_Avg” increased abnormally at the time point “28 June 2016 19:00”. Based on this abnormal phenomenon, troubleshooting was conducted for the abnormal time period of this measurement point.

As shown in Table 2, unit T07 has a TRANSFORMER-type fault recorded at the time point “10 July 2016 3:46”. Among them, the fault diagnosis results of unit T07 at the abnormal time of the “HVTrafo_Phase3_Temp_Avg” measurement point are shown in Figure 11. In the fault diagnosis for the abnormal time period of this measurement point, the fixed-threshold alarm time point is “28 June 2016 23:00” and the adaptive-threshold alarm time point is “28 June 2016 20:00”. Both threshold alarm time points are earlier than the fault recording point, and the adaptive-threshold alarm time is 3.0 h earlier than the fixed-threshold alarm time.

In summary, both fixed and adaptive thresholds advance the alarm ahead of the fault recording time point, as shown in Table 9. However, adaptive-threshold diagnosis has better results in terms of early alarms. Among them, the GENERATOR fault type is 1.5 h ahead, the GENERATOR_BEARING fault type is about 5.8 h ahead, and the TRANSFORMER fault type is 3.0 h ahead. Therefore, it can be seen that the adaptive-threshold diagnosis method proposed in this study has been effectively verified in practical applications.

## 5. Conclusions

In this study, a multi-objective regression method based on SCADA temperature data is proposed, aiming at scientific diagnosis and accurate alarm of wind turbine problems. The method employs the LightGBM, XGBoost, SGDRegressor, and stacking integrated algorithms to construct multiple regression models and perform fault diagnosis by fitting the variation trend of residuals. The Pearson correlation coefficient, Spearman correlation coefficient, and variance selection were adopted to achieve data feature screening, and multiple target temperature features corresponding to the actual needs of different components were selected. To further optimize the model performance, the Bayesian tuning method was introduced in this study to improve the generalization ability and prediction performance of the model. In addition, the 3σ criterion and the temporal sliding window were applied to propose an adaptive-threshold fault precision diagnosis and alarm method. The application of this method for fault diagnosis finds that this aspect achieves good results in the fault diagnosis of three components, i.e., the generator, generator bearing, and transformer, showing certain applicability.

In addition, a WTG is a nonlinear multi-coupled system with complex interactions between different components. Therefore, how to use SCADA data to improve the comprehensive fault diagnosis of multiple components of WTGs is still a topic worthy of in-depth research. Optimization of the model’s performance and improvement of the training speed are the key points of future research. How to further analyze the fault correlation among wind turbine components and how to improve the accuracy and robustness of fault diagnosis methods need to be further explored.

## Figures and Tables

**Figure 1 sensors-23-06198-f001:**
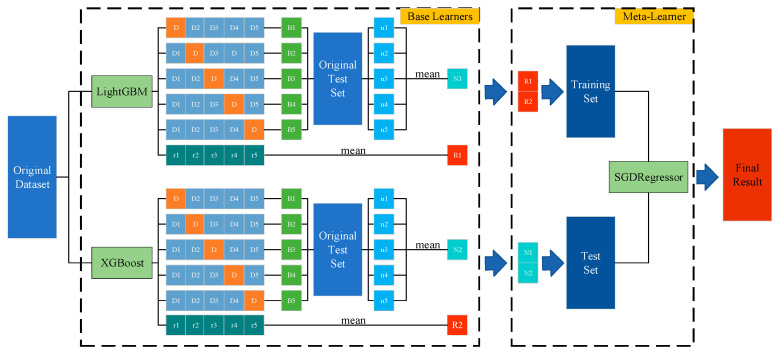
The computational flow of the stacking integration algorithm.

**Figure 2 sensors-23-06198-f002:**
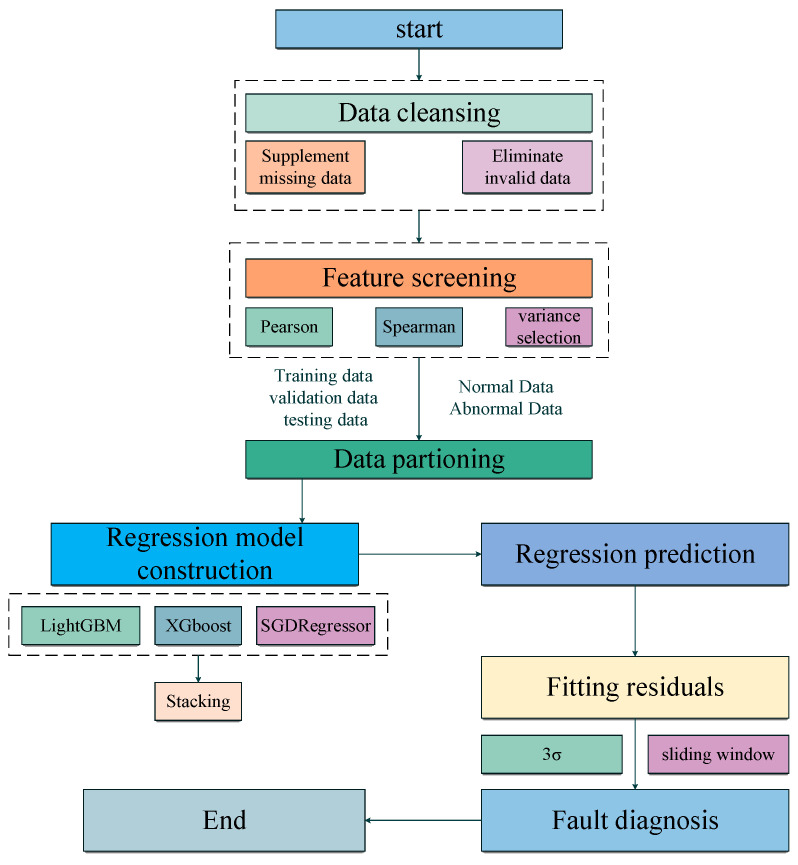
Modeling process of fault diagnosis.

**Figure 3 sensors-23-06198-f003:**
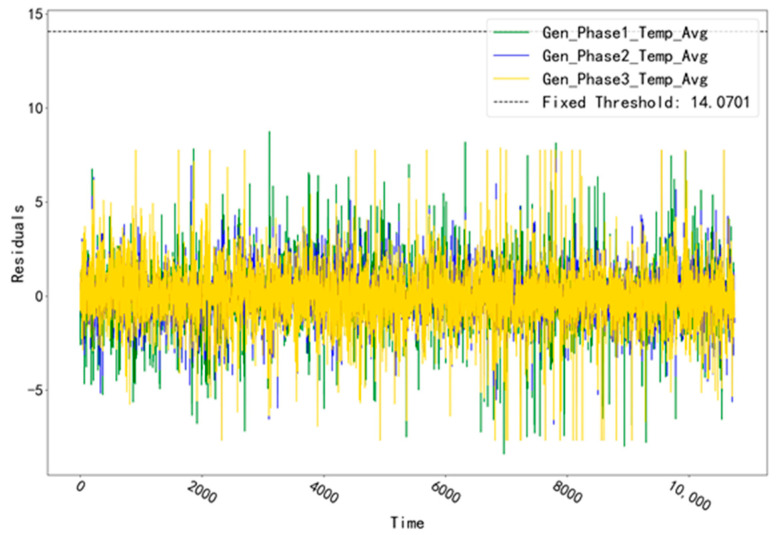
Fitting residual curves for unit T06 for the GENERATOR validation set.

**Figure 4 sensors-23-06198-f004:**
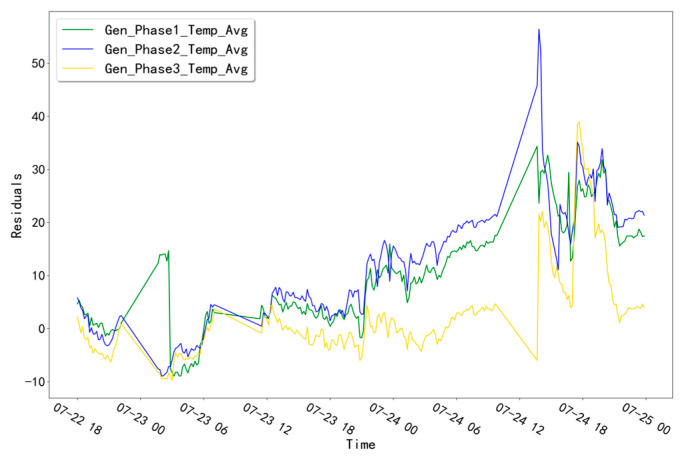
Fitting residual curves for unit T06 for the GENERATOR test set.

**Figure 5 sensors-23-06198-f005:**
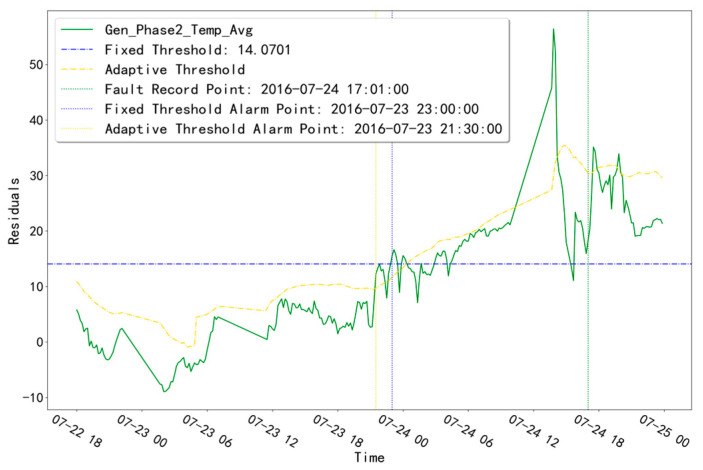
Troubleshooting results of unit T06 in the abnormal time of the “Gen_Phase2_Temp_Avg” measurement point.

**Figure 6 sensors-23-06198-f006:**
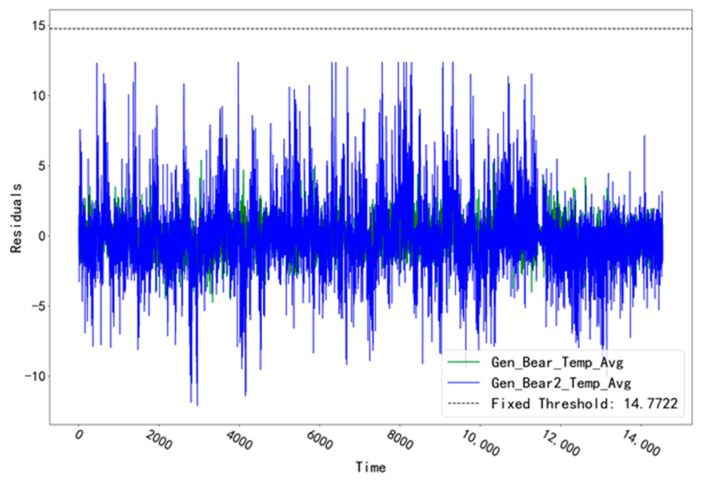
Fitting residual curves for unit T07 for the GENERATOR_BEARING validation set.

**Figure 7 sensors-23-06198-f007:**
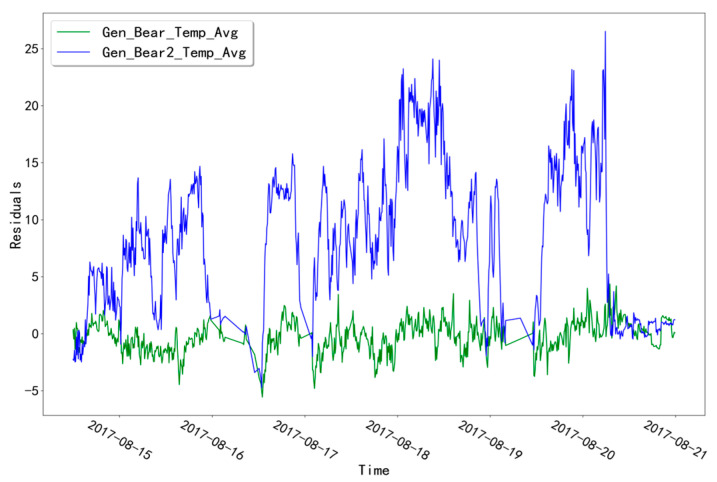
Fitting residual curves for unit T07 for the GENERATOR_BEARING test set.

**Figure 8 sensors-23-06198-f008:**
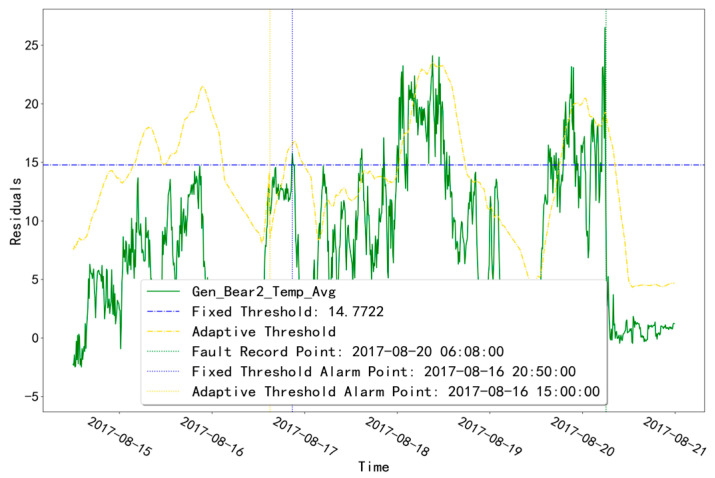
Troubleshooting results of unit T07 in the abnormal time of the GENERATOR_BEARING measurement point.

**Figure 9 sensors-23-06198-f009:**
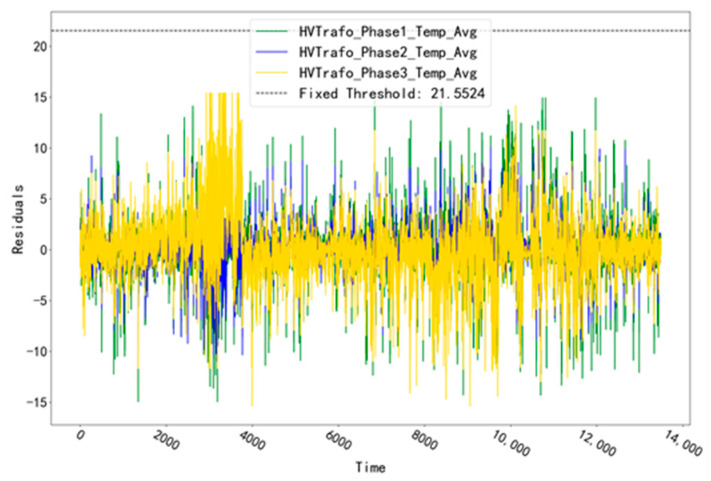
Fitting residual curves for unit T07 for the TRANSFORMER validation set.

**Figure 10 sensors-23-06198-f010:**
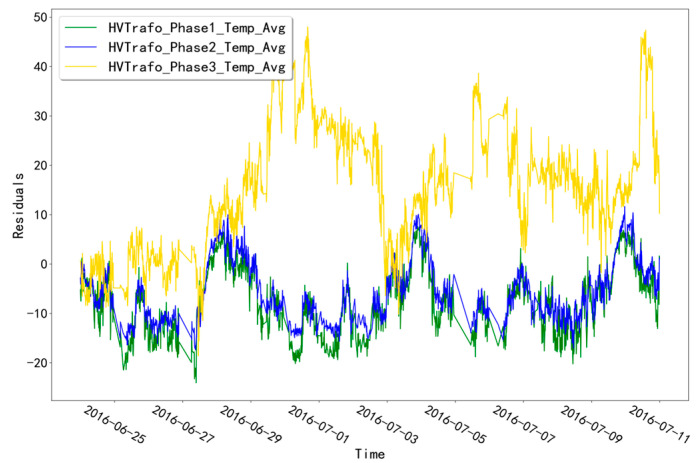
Residual curves for unit T07 for the TRANSFORMER test set.

**Figure 11 sensors-23-06198-f011:**
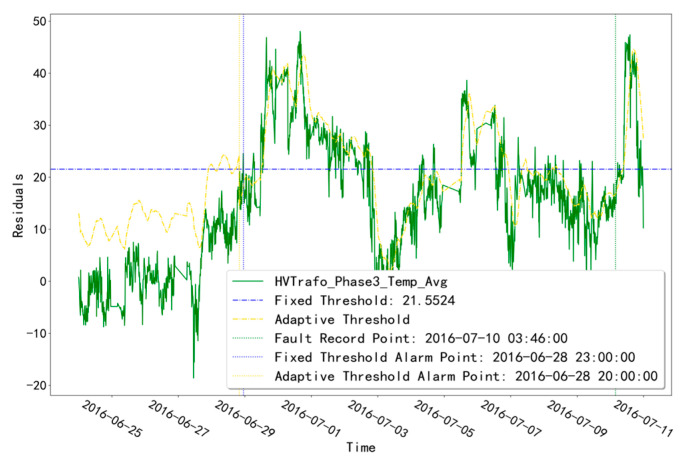
Troubleshooting results of unit T07 for the TRANSFORMER measurement point abnormal time.

**Table 1 sensors-23-06198-t001:** Presentation of selected datasets.

	Timestamp	Gen_RPM_Max	……	Gen_Bear2_Temp_Avg	Nac_Direction_Avg
T11	1 January 2016 00:00	1339.4	……	38	206.9
T06	1 January 2016 00:00	1270.0	……	35	204.6
T01	1 January 2016 00:00	1277.4	……	37	218.5
T09	1 January 2016 00:00	1376.7	……	33	214.0
T07	1 January 2016 00:00	1317.5	……	39	197.3
……	……	……	……	……	……
T11	31 December 2017 23:50	1320.5	……	37	331.5
T07	31 December 2017 23:50	1329.9	……	32	334.8
T01	31 December 2017 23:50	1273.1	……	38	347.8
T09	31 December 2017 23:50	1258.3	……	32	339.7
T06	31 December 2017 23:50	1270.8	……	40	325.1

**Table 2 sensors-23-06198-t002:** Recorded data of faults.

Turbine_ID	Component	Timestamp	Remarks
T06	GENERATOR	24 July 2016 17:01	Generator temperature sensor failure
T07	GENERATOR_BEARING	20 August 2017 06:08	Generator bearings damaged
T07	TRANSFORMER	10 July 2016 03:46	High temperature transformer

**Table 3 sensors-23-06198-t003:** Missing data.

Feature Column	Rate of Missing (%)
Gen_Bear_Temp_Avg	0.001342
Grd_Prod_CosPhi_Avg	0.001342

**Table 4 sensors-23-06198-t004:** Columns of target characteristics.

Types of Faults	Target Characteristics Column
GENERATOR	Gen_Phase1_Temp_Avg, Gen_Phase2_Temp_Avg, Gen_Phase3_Temp_Avg
GENERATOR_BEARING	Gen_Bear_Temp_Avg, Gen_Bear2_Temp_Avg
TRANSFORMER	HVTrafo_Phase1_Temp_Avg, HVTrafo_Phase2_Temp_Avg, HVTrafo_Phase3_Temp_Avg

**Table 5 sensors-23-06198-t005:** Training set, validation set, and test set.

Turbine_ID	Fault Type	Training Set	Validation Set	Test Set
T06	GENERATOR	(42,985, 70)	(10,747, 70)	(18,694, 70)
T07	GENERATOR_BEARING	(58,187, 66)	(14,547, 66)	(14,283, 66)
T07	TRANSFORMER	(53,965, 63)	(13,492, 63)	(13,054, 63)

**Table 6 sensors-23-06198-t006:** Model prediction results for unit T06 for the GENERATOR fault type.

	R^2^			RMSE		
	Train	Validation	Test	Train	Validation	Test
LightGBM	0.9956	0.9920	0.9030	1.1946	1.6048	8.7661
XGBoost	0.9903	0.9887	0.8979	1.7636	1.9091	8.9972
SGDRegressor	0.9707	0.9710	0.9052	3.0659	3.0621	8.6629
Stacking	0.9975	**0.9939**	0.9046	0.8923	**1.4009**	8.6974

**Table 7 sensors-23-06198-t007:** Model prediction results for unit T07 for the GENERATOR_BEARING fault type.

	R^2^			RMSE		
	Train	Validation	Test	Train	Validation	Test
LightGBM	0.9796	0.9682	0.9029	1.8681	2.3297	4.5260
XGBoost	0.9707	0.9646	0.9087	2.2399	2.4564	4.3866
SGDRegressor	0.9179	0.9170	0.9026	3.7477	3.7633	4.5198
Stacking	0.9956	**0.9787**	0.9088	0.8691	**1.9080**	4.3831

**Table 8 sensors-23-06198-t008:** Model prediction results for unit T07 for the TRANSFORMER fault type.

	R^2^			RMSE		
	Train	Validation	Test	Train	Validation	Test
LightGBM	0.9131	0.8903	0.6322	2.9524	3.3021	8.9349
XGBoost	0.9234	0.8997	0.6574	2.7529	3.1548	8.6556
SGDRegressor	0.7566	0.7553	0.6192	4.9123	4.9335	9.0760
Stacking	0.9663	**0.9197**	0.6572	1.8332	**2.8169**	8.6679

**Table 9 sensors-23-06198-t009:** Summary of troubleshooting results.

Turbine_ID	Component	Adaptive Record	Fixed Threshold	Adaptive Threshold	Amount of Lead Time
T06	GENERATOR	24 July 2016 17:01	23 July 2016 23:00	23 July 2016 21:30	1.5 h
T07	GENERATOR_BEARING	20 August 2017 06:08	16 August 2017 20:50	16 August 2017 15:00	5.8 h
T07	TRANSFORMER	10 July 2016 03:46	28 June 2016 23:00	28 June 2016 20:00	3.0 h

## Data Availability

The data used in this study are publicly available and can be accessed from https://opendata.edp.com/ without restrictions. For any additional enquiries regarding the data or requests for collaboration, please contact the corresponding author.

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
