# Peer review of "Fault Diagnosis of Wind Turbine Generators Based on Stacking Integration Algorithm and Adaptive Threshold"

_sensors, 2023, doi:10.3390/s23136198_

Round 1

Reviewer 1 Report

Comments to the Author

Manuscript entitled “Fault Diagnosis of Wind Turbine Generator Based on Stacking Integration Algorithm and Adaptive Threshold.“ The work carried by the author looks interesting. I have certain comments on the manuscript.

1. Details about temperature sensor and their placement can be discussed

2. Please add Abbreviations section. Please ensure that you are using abbreviation consistently.

3. How did the authors select the feature selection? Selection of feature filtering(selection) need to be provided.

4. The classifiers provided are already established while the dataset is acquired from public dataset. How do the authors claim that their work is novel?

5. Additionally, regression analysis was carried but the result plots were not mentioned.

6. The work seems like an application of the regression analysis. How do you convince that this is not a mere application?

7. The claim states that the fault recognition is earlier than the conventional methods. It would be goof if there is a state of the art comparison.

8. Novelty and technical contributions can be listed at the end of the introduction section.

Overall, the manuscript is well-written, and I believe these suggestions will help improve its clarity and readability.

Reviewer 2 Report

In this article, the authors conducted an analysis to invent a new method for fast and accurate fault diagnosis of Wind Turbine Generator (WTG). This study employed a Stacking integration model based on Light Gradient Boosting Machine (LightGBM), eXtreme Gradient Boosting (XGBoost), Stochastic Gradient Descent Regressor (SGDRegressor) using machine learning algorithms based on publicly available datasets from Energias De Portugal.The authors presented a well-organized article that was easy to read and understand. There is a transparent methodology.

Regardless, the reviewer suggests the authors consider the following issues:

·         Line 66: Please explain the term  NDT.

·         Line 75-76: Please give more information about the Supervisory Control and Data Acquisition (SCADA) data system.

·    Line 157, -162: The loss function and the regularization term  (L2parametric) should be further described and explained.

·         Line 170: Please give the definition of overfitting.

·     Line 250-251: The features were filtered by Pearson correlation coefficient, Spearman correlation coefficient and variance. Please be more specific about these indicators.

·         In the Abstract: How does fault diagnosis lag cause high unit operation? Is it not explained at the main text

·         Line 319: What is a quadrature quasi quadrature?

·         Line 39: data ‘’shows’’ correction

·         Line 40: correction -> parameters like wind power energy efficiency, wind turbine fault diagnosis, and fault warning forecasting should…

·   Line 42: What is the difference between regular and post maintenance? Do you mean preventive maintenance?

·         Line 45: Correction -> ‘’Wind turbine generators are’’

·         Line 50:  Suggestion to change the phrase as follows: ‘’ to achieve an efficient and sustainable diagnosis method’’ to enhance the meaning.

·         Line 66: Define NDT as abbreviation

·         Line 67: Correction -> ‘’abnormalities that are present’’

·         Line 72: Correction -> the data

·         Line 77: Correction -> for all available conditions

·    Line 88: Correction -> have difficulty change to -> face certain difficulties

·         Line 105: Correction -> change ‘’chosen’’ to ‘’selected’’

Reviewer 3 Report

The goal of the paper is to address the issue of fault diagnosis in wind turbine generators (WTGs) by proposing a new method that enables accurate and rapid diagnosis. To achieve this goal, the study constructs a Stacking integration model based on three machine learning algorithms: Light Gradient Boosting Machine (LightGBM), eXtreme Gradient Boosting (XGBoost), and Stochastic Gradient Descent Regressor (SGDRegressor). These algorithms are trained using publicly available datasets from Energias De Portugal (EDP). The proposed model is automatically tuned for hyperparameters during training using Bayesian tuning. The evaluation of the model's performance is based on the coefficient of determination (R2) and Root Mean Square Error (RMSE). Additionally, the paper introduces a final adaptive threshold method for accurate fault diagnosis and alarming by calculating the fitted residuals of the test set.

I believe that the work presents relevant results. Below, I present some concerns/recommendations that need to be addressed by authors before publication:

- The flowchart shown in figure 2 is quite poor. The authors should complement it with more details of the proposed process, in order to help the reader in understanding the method.

- The way results are presented needs to be improved. I suggest that the authors add a table with the main results, presenting the main values obtained for each failure mode and each threshold method. This will help the reader to compare the performance achieved between methods and between failure modes.

- Is there a way of calculating the false alarm rates over time? This is an important metric for comparing fault detection and diagnosis methods, but it requires a lot of data to be computed. Authors should calculate such a metric if possible and, if not, comment why it is not possible.

- The authors compared the results between two types of threshold determination, but did not compare the results of the detection method with another existing one. Has this database already been used for this purpose in other articles? If so, the results should be compared.

Minor comments:

- Conclusions is actually chapter 5

- See lines 278 and 279 to correct the paragraph separated in them

Minor editing of English language required

Round 2

Reviewer 1 Report

No further comments